# Modeling of the Spatial Distribution of Forest Carbon Storage in a Tropical/Subtropical Island with Multiple Ecozones

**DOI:** 10.3390/plants12152777

**Published:** 2023-07-26

**Authors:** Ting-Wei Chang, Guan-Fu Chen, Ken-Hui Chang

**Affiliations:** 1Department of Environmental and Life Sciences, University of Shizuoka, 52-1 Yada, Suruga Dist., Shizuoka 422-8526, Japan; twchang@u-shizuoka-ken.ac.jp; 2Department of Safety, Health and Environmental Engineering, National Yunlin University of Science & Technology, 123 University Road, Section 3, Douliu, Yunlin 64002, Taiwan; newman2100@gmail.com

**Keywords:** forest management, forest carbon, carbon sequestration, greenhouse gas (GHG), sustainable forestry, inventory, grid modeling

## Abstract

Visual data on the geographic distribution of carbon storage help policy makers to formulate countermeasures for global warming. However, Taiwan, as an island showing diversity in climate and topography, had lacked valid visual data on the distribution of forest carbon storage between the last two forest surveys (1993–2015). This study established a model to estimate and illustrate the distribution of forest carbon storage. This model uses land use, stand morphology, and carbon conversion coefficient databases accordingly for 51 types of major forests in Taiwan. An estimation in 2006 was conducted and shows an overall carbon storage of 165.65 Mt C, with forest carbon storage per unit area of 71.56 t C ha^−1^, where natural forests and plantations respectively contributed 114.15 Mt C (68.9%) and 51.50 Mt C (31.1%). By assuming no change in land use type, the carbon sequestration from 2006 to 2007 by the 51 forest types was estimated to be 5.21 Mt C yr^−1^ using historical tree growth and mortality rates. The result reflects the reality of the land use status and the events of coverage shifting with time by combining the two forest surveys in Taiwan.

## 1. Introduction

The substantially increased emission of greenhouse gases (GHGs) through human activities has led to higher radiative forcing, resulting in global warming [1]. Although fossil CO_2_ emission slightly decreased from 2019 to 2020, the mean annual fossil CO_2_ emission amount from 2011 to 2020 increased to a record level of 9.5 ± 0.5 Gt C yr^−1^ [2]. The Intergovernmental Panel on Climate Change (IPCC) reported that anthropogenic activities have caused an estimated increase in the global average temperature of approximately 1.0 °C compared with the average pre-industrial era temperature. This value may reach 1.5 °C between 2030 and 2052 if the current emission rates remain unchanged [3]. Warming-induced climate change leads to notable shifts in ecosystems [4,5], increases the risk of extinction [6,7], increases the frequency of wildfires [8,9], and threatens food production [10,11]. Thus, carbon neutralization is needed to mitigate global warming and its consequences. Schrag [12] proposed three strategies to reduce the net emission of CO_2_: reducing the amount of energy the world uses; expanding the use of carbonless/carbon-neutral energy; and capturing the CO_2_ and then storing it (i.e., carbon sequestration). Forests account for over 45% of terrestrial carbon storage [13]; the carbon sequestration rate of global forests, including their soil, has been estimated to be 2.4 ± 0.4 Gt C yr^−1^, making forests the largest terrestrial carbon sink [14]. Due to the significance of the forest ecosystem, reforestation and afforestation play a positive role in the modern carbon sequestrating demand [15]; appropriate management of forests and the use of forest products can further promote the efficiency of carbon sequestration by forests [16,17]. In contrast, deforestation is deemed to be one of the largest sources of anthropogenic carbon emissions nowadays [18]. To optimize the management of forests for carbon sequestration, assessment of forest carbon storage is essential.

The forest carbon storage of Taiwan, an island with a forest coverage rate of over 60%, has been investigated over the past few decades in numerous studies (e.g., [19,20,21,22,23,24]). Most recently, the total forest growing stock in Taiwan was 502 million m^−3^, and the forest carbon storage was 206 Mt C according to the 4th Forest Resources Survey (FRS) in Taiwan conducted from 2008 to 2013 [25]. However, on the other hand, visual data on the geographic distribution of forest carbon storage in Taiwan, which is crucial for publicizing and policymaking, has lacked in the past few decades until 2015. Presently, the distribution map of forest and bamboo carbon storage depicted in the 4th FRS provided by the Forest Bureau is the only available visual information related to forest carbon storage in Taiwan. Since the 3rd FRS, which was finished in 1993 [26], there has been a marked shift in land use and forest distribution due to the promulgation of the Soil and Water Conservation Act in 1993 and the implementation of afforestation incentivization policies aimed at incentivizing afforestation. Thus, the present carbon storage map in the 4th FRS is insufficient to reflect the temporal changes in regional forest carbon storage occurring in Taiwan.

This study aims to establish a grid model capable of output visualized data which uses databases of forest distribution, stock volume equation, ecozone distribution, land use type, biomass–carbon converting coefficients, forest growth rate, and mortality rate to simulate forest carbon storage and sequestration rate. Here, we used 2006 as the representative year, since it is approximately the midway between the 3rd and 4th FRSs.

## 2. Results

### 2.1. Estimation of Forest Carbon Storage

The estimation result of stock volume distribution in Taiwan is shown in Figure 1. The total *V*_stand_ was 499,448,000 m^3^, and the average *V*_A_ was 216 m^3^ ha^−1^. Among them, coniferous and broadleaved forests occupied a value of 481,731,000 m^3^, with an average *V*_A_ of 227 m^3^ ha^−1^; bamboo forests occupied 17,717,000 m^3^, with an average *V*_A_ of 91 m^3^ ha^−1^. Most of the grids with large stock volume were distributed in the mountainous area at the center and the northern-east part of Taiwan, with an altitude of approximately 2500 to 3000 m, despite that some fault zones showed low or no stock. In contrast, plains on the eastern sides did not have notable forest stock, only some sparse distribution at the coastal zone.

The estimation result of stock volume of timber with respect to the 51 forest types is shown in Table 1. The spatial distribution of carbon stock regardless of forest type is demonstrated in Figure 2. The overall carbon storage amount of the 51 forest types was estimated as 165.65 Mt C in Taiwan. Among them, the carbon storage contributed by the natural forests was 114.15 Mt C (68.9%), whereas the plantations contributed 51.50 Mt C (31.1%), with a ratio of about 7:3. The total carbon storage in the broadleaved forests was 86.75 Mt C (52.4%), followed by coniferous forests with 39.65 Mt C (23.9%); mixed forests, excluding bamboo, with 31.97 Mt C (19.3%); and bamboo and mixed bamboo forests with 7.29 Mt C (4.4%). Among all the forest types, natural broadleaved mixed forest (MB-NF) exhibited the largest carbon storage of 51.79 Mt C. The natural coniferous forest types had, on average, higher values in carbon storage per unit area. Among them, natural spruce forest (SPR-NF) had the highest value of 221.48 t C ha^−1^. The range of carbon storage per unit area of pure bamboo forests was between 16–70 t C ha^−1^, but most of them were less than 30 t ha^−1^. Only TG-BAM-NF and TG-BAM-P exhibited higher values—69.59 and 69.99 t C ha^−1^, respectively. 

The distribution of the total carbon storage of the 26 forest types output from Monte Carlo simulation results is shown in Figure 3, and the respective results of each forest type are shown in Table 2. According to the 95% confidence interval, the amount ranged from 110.30 to 158.81 Mt C (−16 to 21%) and was slightly positively skewed. Generally, *R*_97.5_ of the forest types ranged from −75% to –20%; the *R*_2.5_ ranged from 20% to 80%. FIR-NF exhibited the largest uncertainty, where its *R*_97.5_ and *R*_2.5_ were −74% and 78%, respectively. It was followed by the natural tsuga forest (TSU-NF), which exhibited a range from −56% and 80%. CYP-NF has the smallest uncertainty, ranging from −19% to 30%. The range of the broadleaved forest types was generally between −35% to 55%.

### 2.2. Estimation of Annual Carbon Sequestration from 2006 to 2007

∆*C* of each forest type is shown in Table 3. As the result, the total ∆*C* from 2006 to 2007 in Taiwan was estimated to be 5.21 Mt C yr^−1^. And the average ∆*C*_A_ was 2.26 t C ha^−1^ yr^−1^. The natural forests contributed about 2.22 Mt C yr^−1^ (42.6%), while that of the plantations was 2.99 Mt C yr^−1^ (57.4%), with a ratio of about 4:6. In terms of wood type, the ∆*C* of broadleaved forests was 3.46 Mt C yr^−1^ (66.4%), followed by coniferous forests with 0.96 Mt C yr^−1^ (18.4%); other mixed forests with 0.45 Mt C yr^−1^ (8.7%); and bamboo forests and mixed bamboo forests with 0.34 Mt C yr^−1^ (6.5%). Among the forest types, ∆*C* of the private broadleaved plantation (P-B-P) reached 1.73 Mt C yr^−1^, accounting for 41.9% of total forest carbon sequestration in the simulation.

## 3. Discussion

### 3.1. Estimation of Forest Carbon Distribution in Taiwan

Our estimation of carbon storage of Taiwan’s 51 forest types in 2006 showed a total value of 165.6 Mt C, and the forest carbon storage per unit area was 71.56 t C ha^−1^. This is approximately the same as the world average of carbon stored in above-ground and below-ground biomass (72.6 t C ha^−1^) and surpasses the Asian average (60.3 t C ha^−1^) [27]. However, the below-ground carbon of partial species was not included in our estimation, thus, the actual amount is expected to be higher. Evaluation studies on forest carbon storage during 1993 to 2014 in Taiwan usually ignored the bamboo forest and private forests. Here if we exclude the storage of bamboo forests, private forests, and newly added forests from our estimation, the 95% confident range of forest carbon storage is 110.07−159.59 Mt C in 2006. Chiou et al. [22] used the 3rd FRS data and SPOT satellite imagery in 2004 to estimate the carbon storage of state-owned coniferous and broadleaved forests in Taiwan and came out with much higher results than our estimation, where the storage ranges from 146 to 261 Mt C when using the local parameters and ranges from 161 to 286 Mt C when using IPCC recommended parameters. Wang [24], meanwhile, estimated that the carbon storage of state-owned coniferous and broadleaved forests in Taiwan in 2006 was 137.27 Mt C, and the average carbon storage per unit area was 99.52 t ha^−1^, which was similar to the results of this study. Generally, our estimation of forest carbon stock in 2006 is within a reasonable bound to previous evaluation studies.

By simulating the spatial distribution of forest carbon storage, there was a high *C*_A_ in high regions with altitudes of 2500 to 3500 m (123.03 t C ha^−1^), which can be attributed to natural coniferous forest types including CYP-NF, PIN-NF, SPR-NF, and other natural coniferous forest (O-C-NF). This correctly reflected the natural coniferous forests in the central mountains of Taiwan, where the forests show old-growth features and enriched volume stock. Among them, FIR-NF showed much lower *C*_A_ than other natural coniferous types and a large variation in the uncertainty test, which might be not a misestimation because the *DBH* and *H* database used in the simulations are from permanent plots and can therefore be considered close to the real conditions. In fact, the distribution of fir forests is at an altitude range of 3100–3600 m, which is much higher than that of other pure coniferous forest types, and tall fir trees are easily broken due to strong mountain winds, resulting in a smaller wood stock [28,29]. Thus, the larger variation of FIR-NF was more likely due to its nature of wide variability of *DBH* and *H*. In addition, the *C*_A_ of coniferous plantations in the simulation was much smaller than that of natural coniferous forests (57.87 t C ha^−1^). This correctly reflects the status of coniferous plantations, where the stand age is much younger than that of natural forests in Taiwan. The estimated distribution also demonstrates that forest regions at lower altitudes (<1500 m) show generally lower *C*_A_ (64.02 t C ha^−1^). This is reasonable because these regions were mainly occupied by broadleaved forests. These broadleaved forests were mostly secondary forests, which have lower stock volumes compared to coniferous forests, although the lower-*C*_A_ broadleaved forests occupied three times more total coverage area than the coniferous forests did. Thus, our simulation showed a larger total carbon storage of the broadleaved forest types than that of the coniferous forest types. However, the simulated carbon storage of broadleaved forests was mostly contributed by MB-NF and P-B-P, whose composition of species were not specifically defined, and the parameters were roughly defined with the averaged value of other corresponding forest types. Thus, our simulation may only reflect the characteristics of carbon storage distribution of broadleaved forests in a low resolution. Also, on the other hand, the low *C*_A_ of JE-P in the simulation was attributed to the low *H* and *N* in the database, which reflected the typical wide crown and short stem of Japanese elm in reality.

Although bamboo forests only contribute a low proportion of forest coverage, in the context of global warming, the habitat of coniferous forests may decrease, and bamboo forests may expand to replace the coniferous forests according to vegetation habitat simulations [30,31]. With regard to their increasing importance, we included the bamboo forests and bamboo-mixed forests in our estimation. It is demonstrated that the low *C*_A_ and the low coverage area result in a contribution of less than 5% of the forest’s carbon storage. In the calculation of most bamboo species, we used the parameters of *P. makinoi*. However, the rhizome system of bamboo species can be divided into two types: pachymorph (clumped) and leptomorph (scattered) [32]. The shape of the culm and the distribution pattern in a plot are considerably different between the two types; thus, it may not correctly reflect the actual growth of the clump bamboo forest. Our result is very close to that of the simulation by Lin [23], in which bamboo and mixed bamboo forests had a carbon storage of 6 Mt C. A more recent study by Yen et al. [33] estimated the bamboo carbon storage per unit area with a value of 49.81 t C ha^−1^, which was much higher than our result. Due to the discrepancy, further investigation on the carbon-storing characteristics of bamboo species is suggested to give a more definite inventory of bamboo carbon storage in Taiwan.

### 3.2. Carbon Sequestration Simulation Results

The forest stock volume of Taiwan has constantly increased from 1993 to 2015 [25]. According to the 3rd FRS, the total forest stock volume of coniferous and broadleaved forests was estimated to be 358,209,000 m^3^, and the stock volume per unit area was estimated to be 184 m^3^ ha^−1^ in 1993 [26]. The 4th FRS reported that in 2015, the total stock volume of coniferous and broad-leaved forest was 494,016,000 m^3^, and the volume per unit area was approximately 255 m^3^ ha^−1^. Differentiating between those of the two FRS, we can see that the total stock volume increased by approximately 135,807,000 m^3^, with an average annual increase rate of 6,173,000 m^3^. Thus, remarkable carbon sequestrations by forest ecosystems in Taiwan are expected. As the supplementary between the two FRS, we estimated the total stock volume of coniferous and broad-leaved forests to be 411,349,000 m^3^ and 255 m^3^ ha^−1^ in 2006. From 1993 to 2006, the total stock volume increased by 53,140,000 m^3^, and the stock volume per unit area increased by 72 m^3^ ha^−1^. On average, the stock volume increased by 4,088,000 m^3^ per year, and the stock volume per unit area increased by 5.5 m^3^ ha^−1^ per year. From 2006 to 2015, the total stock volume increased by approximately 82,667,000 m^3^, with an average increase rate of 9,185,000 m^3^ per year, which is more than two-fold that of the value from 2006 to 2015. The stock volume per unit area had no substantial change from 2006 to 2015 according to the estimations, which infers a considerable increase in forest coverage area between 2006 and 2015, and this is consistent with the period of afforestation incentivization of Council of Agriculture. 

We estimated the annual carbon sequestration of forests from 2006 to 2007, by assuming no significant change in vegetation distribution, and the change in forest biomass is only attributed to natural death and growth during the period. This assumption is reasonable because timber harvesting was strictly limited by law in Taiwan during the period [34], and large-area forest fires rarely occurred [35]. According to the results, it can be seen that broadleaved forest types accounted for most (3.64 Mt C y^r−1^, 66.4%) of the total carbon sequestration amount (5.21 Mt C yr^−1^) in Taiwan’s forests. Private forest types, including P-B-P, P-C-P, P-BAM-P, and private coniferous and broadleaved mixed plantation (P-M-CB-P) account for a considerable amount of carbon sequestration, while they were usually omitted in previous studies. If their contributions to carbon sequestration are deducted, the total carbon sequestration of state-owned forests in Taiwan was 3.44 Mt C yr^−1^. This value is lower than those found by Wang [19], Lin et al. [21], and Chiou [22], in which annual carbon sequestrated by forests were estimated as 4.74, 4.56, and 3.65–6.90 Mt C yr^−1^, respectively. The main differences in the method of our estimation to other studies are the land use database, the parameters of growth and mortality rate, or the parameters related to biomass–carbon conversion. Where Wang [19] used the growth rate conducted by 2nd FRS, and calculation and biomass conversion parameters recommended in the Revised 1996 IPCC Guidelines for National Greenhouse Gas Inventories [36], Lin et al. [21] used the land use database conducted by the 3rd FRS [26], and simulated the carbon storage with own-conducted biomass–carbon conversion parameters; Chiou et al. [22], meanwhile, used the local parameters and IPCC-recommended parameters to estimate the total carbon sequestration from 1995 to 2006, with consideration of land use change (3.65–6.16 Mt C yr^−1^ and 4.15–6.90 Mt C yr^−1^, respectively). According to the 4th FRS, forest carbon sequestration in Taiwan was 4.6−5.7 Mt C yr^−1^ from 1993 to 2013, which is close to values reported in other studies and higher than the estimated value of this study. The major reason for the underestimation of this study is the omission of below-ground biomass in the database. Also, the growth rate mortality rate of trees which were conducted in the past may not conform to the actual status of the stands in 2006. Additionally, the forest distribution had been largely changed since 2006, and our result may not meet the current reality of forestry in Taiwan. Thus, an update on land use, biomass–carbon conversion, growth rate, and mortality rate databases is suggested for a more accurate estimation that fits the current situation in Taiwan.

## 4. Methodology

### 4.1. Specification of the Grid Carbon Storage and Carbon Sequestration Modeling of Taiwan

The target region is the main island of Taiwan. Taiwan is located at longitude 124° E to 119° E and latitude 21° N to 25° N. To the west of the land is the Taiwan Strait, and to the east lies the Pacific Ocean. The total area is approximately 36,000 square kilometers. The main climate is tropical and subtropical, with an average annual temperature of 22.0 °C and an average annual rainfall of 2000 mm. The terrain mainly consists of mountains and hills. The highest mountain range is almost 4000 m above sea level, and the plains account for about 1/3 of the island. Since 1993, forest coverage has exceeded 2.1 million hectares (58% coverage) and continues to increase. Due to the influence of vertical changes in climate, there are many species of forest trees, including tropical, temperate, and boreal species. According to Wang et al. [37], there are at least 210 species of tree from 145 genera and 61 families in Taiwan. Based on the database of land use patterns, the delineation of ecozones (specified in Section 2.2) has been added in this study to obtain a map of land use distribution and area information. In addition, the stock volume was calculated according to the volume correlation coefficient unique to each land use type, and then the carbon storage amount was estimated using the biological carbon conversion correlation coefficient. Finally, the map and calculation results were combined to obtain the accumulation and carbon storage distribution of the whole grid.

The 2006 forest distribution used in this study is based on the 3rd FRS, CTCI database and was updated with satellite telemetry data from the Center for Space and Remote Sensing Research of National Central University (CSRSR) from 2005 to 2006 [38]. In terms of the map resolution, each grid area is 1 square kilometer, and each grid land use type area is accurate to within 1 square meter. In the calculation process, the diameter at breast height (DBH, cm) and stem height (H, m) of each forest type, as well as the number of plants in it were entered, and then the appropriate ecozone database for the forest types in the grid was selected according to the ecozone. The appropriate stock volume equation and the conversion coefficient for the forest types were then selected. They were used to calculate the carbon storage per unit area, which was then multiplied by the occupied areas of the corresponding forest types in the grid to calculate the carbon storage of the forest type in the grid. After all of the grids had been estimated, they were combined to obtain the spatial distribution map of carbon storage over Taiwan island.

### 4.2. Algorithms and Parameters

This study determined 51 forest types based on the Taiwan Ecosystem Process Model (TEPM) land-use database [39,40]. The forest types and their abbreviations are listed in Table 4. The guidelines of IPCC suggest that when estimating forest carbon storage, it should be distinguished to accommodate the different growth conditions of different ecozones [41]. In this study, Taiwan was divided into five ecozones, namely subtropical mountain system, subtropical humid forest, tropical rainforest, tropical dry forest, and tropical moist deciduous. The spatial distribution is shown in Figure 4.

The general-purpose forest living carbon storage formula suggested in the 2019 Refinement to the 2006 IPCC Guidelines for National Greenhouse Gas Inventories [40] is as follows:*C* = *A* × *V*_stand_ × *BCEF* × (1 + *R*) × *CF*(1)
where *C* is the mass of carbon contained in biomass (Mt C), *A* is the coverage area (ha), *V*_stand_ is the growing stock volume of tree trunk (m^3^ ha^−1^), *BCEF* is the biomass conversion and expansion factor for expansion of growing stock volume to above-ground biomass (t C m^−3^), *R* is the ratio of below-ground biomass to above-ground biomass (t t^−1^), and *CF* is the carbon fraction of dry biomass (t C t^−1^).

However, at present, database of *BCEF* and assessments of below-ground variables are not adequate in Taiwan. Therefore, in most of this research, only the above-ground biomass is calculated. The below-ground biomass is only estimated for some species. The calculation formula adopts the estimation formula of Lee et al. [42]:*C* = *A* × *V*_stand_ × *EF* × *D* × *CF*(2)
where *EF* is the above-ground biological expansion factor (unit-free), *D* is the basic wood density (t m^−3^), and *CF* is the carbon fraction. Carbon storage per unit area (*C*_A_, t C ha^−1^) is defined as follows:*C*_A_*= C* ÷ *A* × 10^6^ = *V*_stand_ × *EF* × *D* × *CF* × 10^6^(3)

*V*_stand_ is calculated as follows:*V*_stand_ = *N* × *V*_stem_ (*DBH*, *H*)(4)
where *N* (stem ha^−1^) is the stand density of each forest type; *V*_stem_ (m^3^ stem^−1^) is the growing stock volume, which is calculated from *DBH,* and *H* with the stock volume equation of each forest type. 

*EF*, *D*, *CF*, and the volume formulas used for each forest type in this study use the corresponding parameters to the referred plant species shown in Appendix A. Forest types which are comprised of multiple referred plant species use parameters multiplied by the respective weights of each species. Stock volume equations of each plant species at different ecozones are shown in Appendix A. The cypress plantation (CYP-P) takes *Chamaecyparis taiwanensis* as the reference species because it is the main afforestation cypress species in Taiwan. On the other hand, the natural cypress forest (CYP-NF) is mainly a mixture of *C. taiwanensis* and *Chamaecyparis formosensis*, and, thus, CYP-NF is assigned a weight of 0.5 each for the two species. Regarding the pine species, *Pinus massoniana* was taken as the reference species of the pine plantation (PIN-P) because it is introduced as the main afforestation pine species in Taiwan; *Pinus taiwanensis* was taken as the reference species of the natural pine forest (PIN-NF). For forest composition, natural coniferous and broadleaved mixed forest (M-CB-NF) is mainly composed of coniferous and broadleaved forest, with coniferous tree species predominating, including *Chamaecyparis* spp. Therefore, the coniferous forest part of this forest is mixed with other tree species, including needles, red cypress, cypress, and other broad-leaved trees, each with a weight of 0.25. In the forest types with no specific species, such as other natural coniferous forest (O-C-NF), natural coniferous and broadleaved mixed forest (M-CB-NF), natural mixed broadleaved forest (MB-NF), natural broadleaved forest (B-NF), other conifer plantation (O-C-P), mixed conifer plantation (MC-P), coniferous and broadleaved mixed plantation (M-CB-P), mixed broadleaved plantation (MB-P), and private plantations (i.e., P-C-P, P-B-P, P-BAM-P, and P-M-CB-P), their reference species was assigned as “Mixed Conifer” or “Mixed Broadleaf”, whose *D* and *CF* were the averaged value of those of all corresponding species. *D* and *CF* of *Abies kawakamii* and *C. taiwanensis* were the averaged value of those of all the coniferous species in Lin et al. [43]; *D* and *CF* of *Liquidambar formosana* and *Paulownia kawakamii* were the averaged value of those of all the broadleaved species in Lin et al. [43]. All bamboo forest types, *EF*, *D*, *CF,* and volume formulas are all based on the data of *Phyllostachys makinoi* [44].

*DBH*, *H*, *N,* and *A* of each forest type are demonstrated in Appendix A. These parameters are mainly obtained from the database of permanent forest plots in 2001. The *DBH, H,* and *N* of most of the bamboo forest types were obtained from data for *P. makinoi* except *Phyllostachys pubescens* (moso bamboo) forest types (i.e., MOS-BAM-NF and MOS-BAM-P) and *Dendrocalamus latiflorus* (Taiwan giant bamboo) forests (i.e., TG-BAM-NF and TG-BAM-P). For mixed forest types that include bamboo forest, the weight of bamboo is 0.5, and the weight of the other species evenly shares the other 0.5. In this case, the *DBH* and *H* of the other species were the averaged data of the corresponding forest types.

In this study, we estimated the carbon sequestration for one year, from 2006 to 2007. Assuming that there was no significant change in forest area, or biomass losses caused by thinning, pruning, herbivory, diseases, fires, and other disturbance from 2006 to 2007, the forest carbon storage in 2007 can be estimated based on the annual growth rate and mortality rate, and the change in forest carbon storage between the two years is the annual carbon sequestration (∆*C*, Mt C yr^−1^). The estimation formula is as follows:∆*C* = *C*_next_ − *C*(5)
where *C*_next_ (Mt C) is the carbon storage in the next year. It is defined as follows:*C*_next_ = *A* × *V*_stand_ × (1 + *G*) × (1 − *M*) × *BEF* × *D* × *CF*(6)
where *G* (%) is the growth rate of the growing stock volume, and *M* (%) is the mortality rate of trees in a stand. The annual growth and death rate of each forest type were obtained from a survey on the growth and death of forest resources in Taiwan [45] (Appendix A). According to Equations (2), (4) and (5), we can obtain:∆*C* = *C* × (1 + *G*) × (1 − *M*)(7)

The carbon sequestration per unit area (∆*C*_A_, t C ha^−1^ yr^−1^) is calculated as follows:∆*C*_A_ = ∆*C* ÷ *A* × (1 + *G*) × (1 − *M*)(8)

### 4.3. Model Uncertainty Analysis

This study uses the Monte Carlo method to estimate the uncertainty of the carbon storage modeling caused by the variation in *DBH* and *H* of the 26 state-owned forest types (i.e., FIR-NF, TSU-NF, CYP-NF, PIN-NF, SPR-NF, O-C-NF, CYP-P, PIN-P, LF-P, TAI-P, JC-P, TIC-P, O-C-P, MC-P, B-NF, MB-NF, ACA-P, SG-P, CAM-P, ASH-P, JE-P, SDT-P, O-B-P, MB-P, M-CB-NF, M-CB-P), whereas those of the newly added forest types (i.e., NEW-C-P, and NEW-B-P), private forest types, bamboo forest types, and mixed forest types with bamboo remained fixed. First, we established the normal distribution of *DBH* and *H* based on the standard deviation of the *DBH* and *H* database for each forest type. Then, we simulated *DBH* and *H* based on the established normal distribution (regarding *DBH* and *H* are independent of each other) and calculated the carbon storage, repeating 10,000 times. Finally, the two-tailed 95% confidence interval for the 10,000 simulation results, that is, the upper 2.5th (*C*_2.5_, Mt C) and 97.5th percentile (*C*_97.5_, Mt C) values, were used as the error range. The relative carbon storage of the 2.5th (*R*_2.5_, %) and 97.5th percentile (*R*_97.5_, %) was defined as follows:*R*_2.5_ = [(*C*_2.5_ − *C*)/*C*] × 100%(9)
*R*_97.5_ = [(*C*_97.5_ − *C*)/*C*] × 100%(10)

## 5. Conclusions

This study established an estimation method that can demonstrate the distribution of forest carbon storage in Taiwan, which can facilitate better legibility for the public. Different stock volume equations specific to each ecozone within a species were adopted to reflect the morphological variations of tree species in response to climate. Our estimation in 2006 encompassed carbon storage assessments for coniferous, broadleaved, and bamboo forests that are natural and planted. Overall, we estimated the total carbon storage amount of the 51 forest types to be 165.65 Mt C. Our geographic estimation demonstrated that the coniferous forests in the middle and high altitudes have high-density carbon storage, while the broadleaved forests at the middle to low altitudes contribute more than 50% of the carbon storage in forests of Taiwan due to their wide distribution. This estimation reasonably reproduced the forest carbon storage distribution in Taiwan in 2006. A temporal update of the accompanying data also reveals visual temporal–spatial changes in carbon storage between 2006 and 2007. The annual carbon sequestration by the 51 forest types from 2006 to 2007 in Taiwan was estimated to be 5.21 Mt C yr^−1^. In general, this method offers a comprehensive visual depiction of the spatial distribution of forest carbon storage in Taiwan, which is crucial for managing natural or semi-natural forest ecosystems towards more effective carbon sequestration and sustainable forestry practices. Moreover, the visualized information facilitates communication between scientific research and the general public, contributing to the facilitation of policy-making.

## Figures and Tables

**Figure 1 plants-12-02777-f001:**
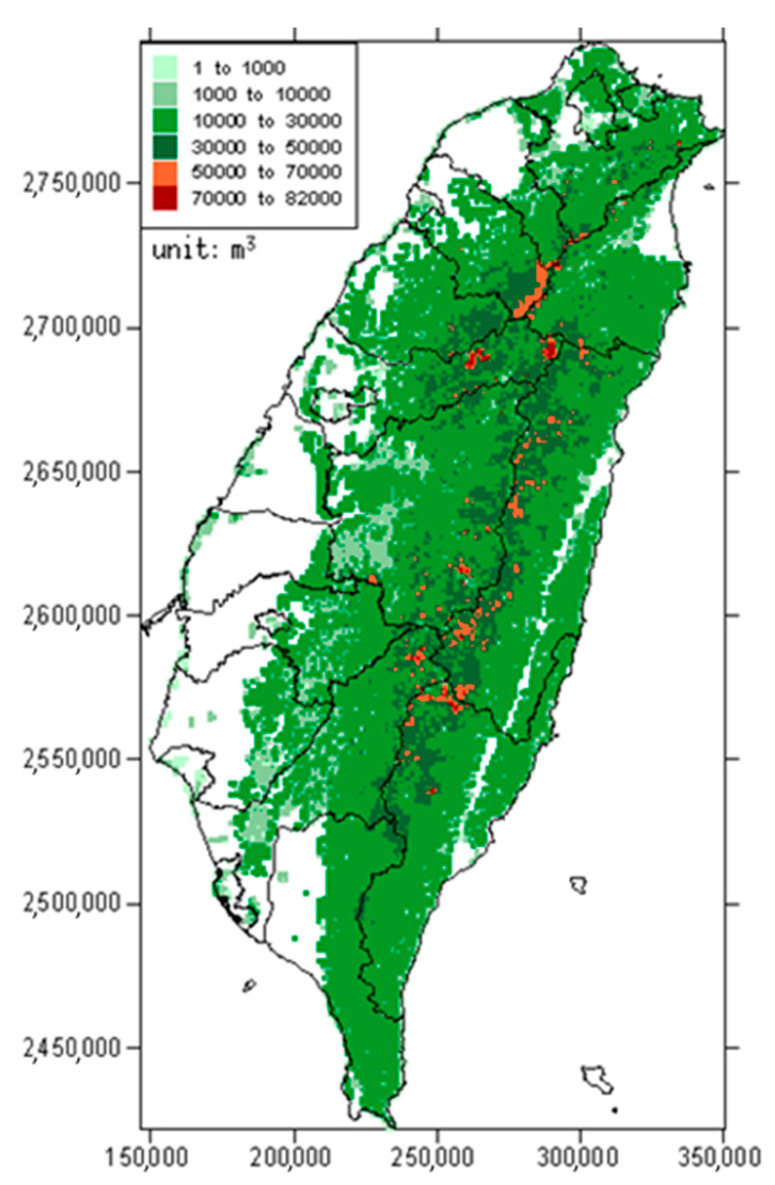
Spatial distribution of *V*_stand_ according to the 51 forest types in Taiwan. Different colors represent different class intervals of stock volume in each grid.

**Figure 2 plants-12-02777-f002:**
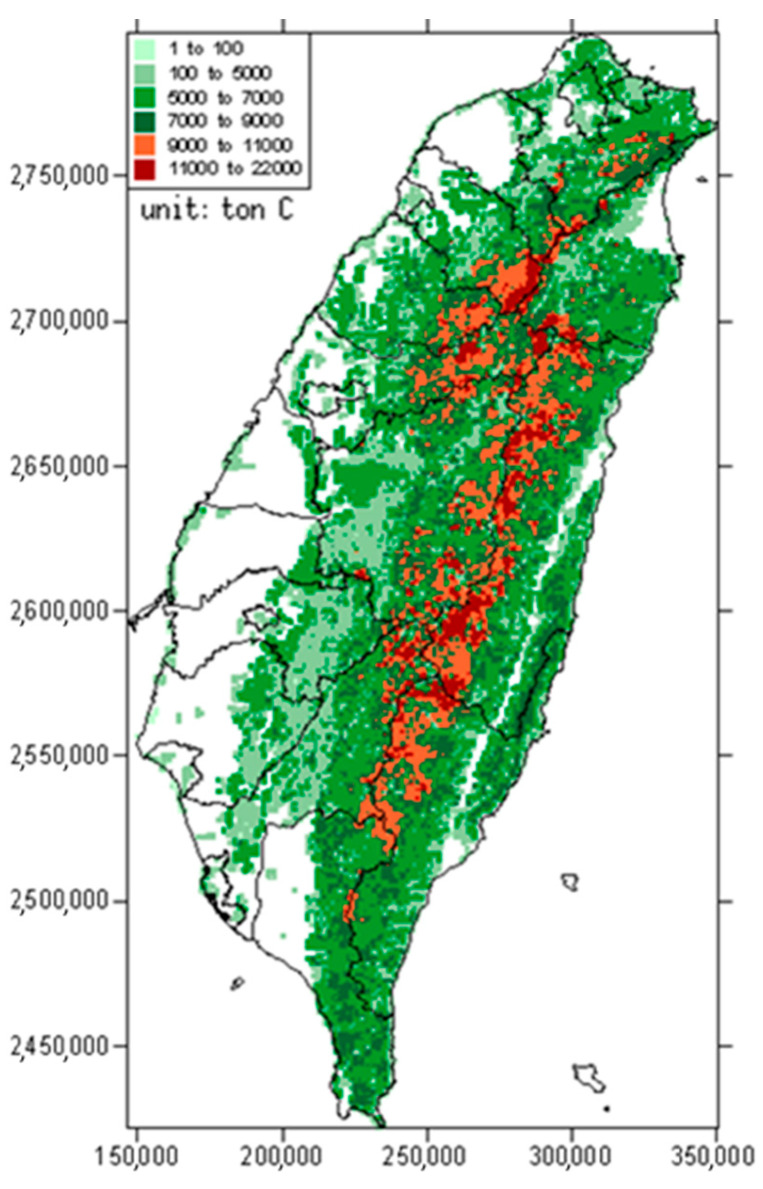
Spatial distribution of forest carbon stock in Taiwan. Different colors represent different class intervals of carbon storage in each grid.

**Figure 3 plants-12-02777-f003:**
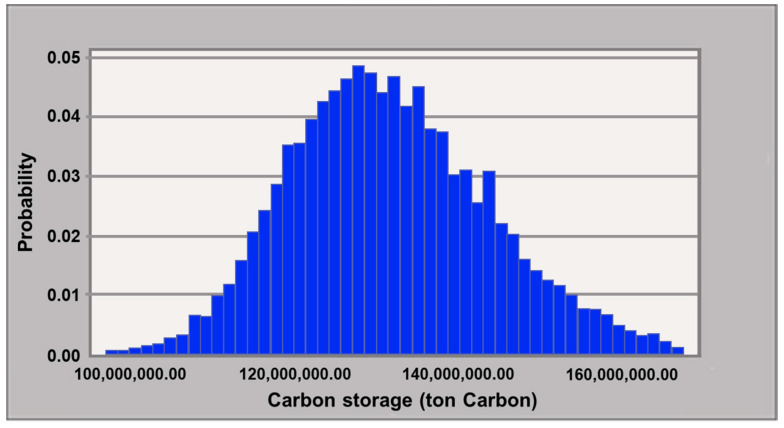
Frequency distribution of the total carbon storage of 26 forest types (*n* = 10,000).

**Figure 4 plants-12-02777-f004:**
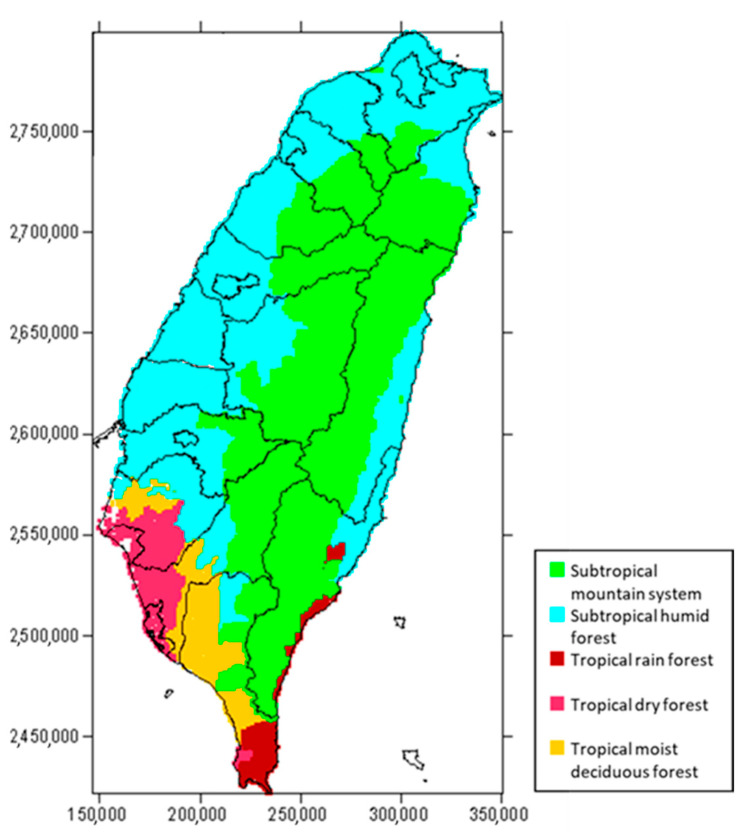
Distribution of the ecozones of Taiwan.

**Table 1 plants-12-02777-t001:** Carbon storage amount per area (*C*_A_), carbon storage amount (*C*), and proportion to the total of 51 forest types.

Wood Type	Regeneration	Forest Type	*C*_A_ (t ha^−1^)	*C* (Mt C)	Proportion to Total (%)
Coniferous	Nature	FIR-NF	47.17	1.01	0.6
		TSU-NF	98.94	5.34	3.2
		CYP-NF	149.94	7.39	4.5
		PIN-NF	107.41	7.55	4.6
		SPR-NF	221.48	1.57	0.9
		O-C-NF	210.12	4.87	2.9
		Average or Total	123.03	27.73	16.7
	Plantation	CYP-P	34.97	0.88	0.5
		PIN-P	67.28	3.11	1.9
		LF-P	74.08	2.24	1.4
		TAI-P	43.56	0.22	0.1
		JC-P	37.79	1.78	1.1
		TIC-P	78.38	0.10	0.1
		O-C-P	115.97	0.20	0.1
		MC-P	67.85	3.05	1.8
		P-C-P	79.64	0.32	0.2
		NEW-C-P	78.52	0.02	<0.05
		Average or Total	57.87	11.92	7.2
	Average or Total		91.92	39.65	23.9
Broadleaved	Nature	B-NF	51.78	0.81	0.5
		MB-NF	71.80	51.79	31.3
		Average or Total	71.38	52.60	31.8
	Plantation	ACA-P	81.06	2.97	1.8
		SG-P	35.61	0.15	0.1
		CAM-P	34.42	0.20	0.1
		ASH-P	54.04	0.72	0.4
		JE-P	7.10	0.04	<0.05
		SDT-P	53.88	0.45	0.3
		O-B-P	66.31	0.92	0.6
		MB-P	59.31	1.55	0.9
		P-B-P	53.88	26.98	16.3
		NEW-B-P	50.53	0.17	0.1
		Average or Total	55.26	34.15	20.6
	Average or Total		64.02	86.75	52.4
Bamboo	Nature	MAK-BAM-NF	18.82	0.19	0.1
		MOS-BAM-NF	26.77	0.05	<0.05
		TG-BAM-NF	69.59	0.72	0.4
		THO-BAM-NF	18.67	0.04	0.0
		BAM-BAM-NF	20.36	<0.005	<0.05
		O-BAM-NF	12.96	<0.005	<0.05
		FAR-BAM-NF	18.73	0.34	0.2
		Average or Total	31.33	1.36	0.8
	Plantation	MAK-BAM-P	18.85	0.15	0.1
		MOS-BAM-P	26.29	0.08	<0.05
		TG-BAM-P	69.99	0.52	0.3
		THO-BAM-P	18.70	0.22	0.1
		BAM-BAM-P	19.66	0.02	<0.05
		O-BAM-P	16.52	0.01	<0.05
		P-BAM-P	18.72	0.41	0.2
		Average or Total	26.24	1.41	0.8
	Average or Total		28.51	2.76	1.7
Mixed	Nature	M-CB-NF	100.20	29.47	17.8
		M-BAMC-NF	126.17	0.01	<0.05
		M-BAMB-NF	45.86	2.92	1.8
		M-BAMCB-NF	74.86	0.06	<0.05
		Average or Total	90.50	32.46	19.6
	Plantation	M-CB-P	62.99	2.49	1.5
		M-BAMC-P	48.04	0.10	0.1
		M-BAMB-P	44.55	1.13	0.7
		M-BAMCB-P	55.43	0.30	0.2
		P-M-CB-P	62.93	0.01	<0.05
		Average or Total	55.53	4.03	2.4
	Average or Total		84.62	36.49	22.0
Average or Total			54.86	165.65	100.0

Average or Total: averaged *C*_A_ is shown; summed *C* is shown; summed proportion to total is shown.

**Table 2 plants-12-02777-t002:** Carbon storage and relative carbon storage of 26 forest types at 97.5th and 2.5th percentage in Monte Carlo simulation (*n* = 10,000).

Forest Type	Carbon Storage (Mt C)		Relative Carbon Storage (%)	
	97.5th	2.5th	97.5th	2.5th
FIR-NF	0.26	1.79	−74	78
TSU-NF	2.33	9.62	−56	80
CYP-NF	6.01	9.60	−19	30
PIN-NF	5.22	10.75	−31	42
SPR-NF	0.73	2.69	−54	72
O-C-NF	3.20	7.19	−34	48
CYP-P	0.60	1.26	−32	43
PIN-P	2.06	4.58	−34	47
LF-P	1.47	3.34	−34	49
TAI-P	0.14	0.32	−35	48
JC-P	1.15	2.64	−35	48
TIC-P	0.07	0.15	−33	51
O-C-P	0.13	0.29	−35	45
MC-P	1.63	3.70	−46	21
B-NF	0.52	1.21	−35	50
MB-NF	33.18	77.95	−36	51
ACA-P	2.11	4.04	−29	36
SG-P	0.10	0.21	−36	43
CAM-P	0.13	0.29	−34	47
ASH-P	0.48	1.06	−34	47
JE-P	0.03	0.06	−34	47
SDT-P	0.29	0.67	−36	49
O-B-P	0.59	1.39	−35	51
MB-P	1.00	2.34	−35	51
M-CB-NF	22.31	40.10	−24	36
M-CB-P	1.63	3.70	−34	49
Total	110.07	159.59	−16	22

**Table 3 plants-12-02777-t003:** Annual carbon sequestration amount per area (Δ*C*_A_), annual carbon sequestration amount (Δ*C*), and proportion to total of 51 forest types.

Wood Type	Regeneration	Forest Type	Δ*C*_A_ (t C ha^−1^ yr^−1^)	Δ*C* (Mt C yr^−1^)	Proportion to Total (%)
Coniferous	Nature	FIR-NF	0.40	0.01	0.2
		TSU-NF	0.84	0.05	0.9
		CYP-NF	1.33	0.07	1.3
		PIN-NF	1.28	0.09	1.7
		SPR-NF	1.88	0.01	0.3
		O-C-NF	4.59	0.11	2.0
		Average or Total	0.68	0.33	6.3
	Plantation	CYP-P	1.85	0.05	0.9
		PIN-P	3.56	0.16	3.2
		LF-P	3.91	0.12	2.3
		TAI-P	2.33	0.01	0.2
		JC-P	2.00	0.09	1.8
		TIC-P	4.07	0.01	0.1
		O-C-P	6.22	0.01	0.2
		MC-P	3.59	0.16	3.1
		P-C-P	4.23	0.02	0.3
		NEW-C-P	4.15	<0.005	<0.05
		Average or Total	0.33	0.63	12.1
	Average or Total		0.45	0.96	18.4
Broadleaved	Nature	B-NF	1.22	0.02	0.4
		MB-NF	1.74	1.26	24.1
		Average or Total	0.58	1.27	24.5
	Plantation	ACA-P	5.19	0.19	3.6
		SG-P	2.34	0.01	0.2
		CAM-P	2.20	0.01	0.2
		ASH-P	3.46	0.05	0.9
		JE-P	0.46	<0.005	<0.05
		SDT-P	3.47	0.03	0.6
		O-B-P	4.23	0.06	1.1
		MB-P	3.78	0.10	1.9
		P-B-P	3.44	1.72	33.1
		NEW-B-P	3.23	0.01	0.2
		Average or Total	0.28	2.18	41.9
	Average or Total		0.39	3.46	66.4
Bamboo	Nature	MAK-BAM-NF	0.98	0.01	0.2
		MOS-BAM-NF	1.39	0.00	0.1
		TG-BAM-NF	3.62	0.04	0.7
		THO-BAM-NF	0.97	<0.005	<0.05
		BAM-BAM-NF	1.04	<0.005	<0.05
		O-BAM-NF	0.68	<0.005	<0.05
		FAR-BAM-NF	0.97	0.02	0.3
		Average or Total	0.61	0.07	1.4
	Plantation	MAK-BAM-P	0.98	0.01	0.2
		MOS-BAM-P	1.37	0.00	0.1
		TG-BAM-P	3.64	0.03	0.5
		THO-BAM-P	0.97	0.01	0.2
		BAM-BAM-P	1.02	<0.005	<0.05
		O-BAM-P	0.86	<0.005	<0.05
		P-BAM-P	0.97	0.02	0.4
		Average or Total	0.73	0.07	1.4
	Average or Total		0.68	0.14	2.8
Mixed	Nature	M-CB-NF	1.42	0.42	8.0
		M-BAMC-NF	5.45	<0.005	<0.05
		M-BAMB-NF	1.98	0.13	2.4
		M-BAMCB-NF	3.24	<0.005	<0.05
		Average or Total	0.66	0.55	10.5
	Plantation	M-CB-P	0.89	0.04	0.7
		M-BAMC-P	2.08	<0.005	0.1
		M-BAMB-P	1.93	0.05	0.9
		M-BAMCB-P	2.40	0.01	0.2
		P-M-CB-P	0.89	<0.005	<0.005
		Average or Total	0.71	0.10	2.0
	Average or Total		0.66	0.65	12.5
Average or Total			0.44	5.21	100.0

Average or Total: averaged Δ*C*_A_ is shown; summed Δ*C* is shown; summed proportion to total is shown.

**Table 4 plants-12-02777-t004:** Abbreviations and areas of the 51 forest types.

Wood Type	Regeneration	Forest Type	Abbreviation
Coniferous	Nature	Natural Fir Forest	FIR-NF
		Natural Tsuga Forest	TSU-NF
		Natural Cypress Forest	CYP-NF
		Natural Pine Forest	PIN-NF
		Natural Spruce Forest	SPR-NF
		Other Natural Coniferous Forest	O-C-NF
	Plantation	Cypress Plantation	CYP-P
		Pine Plantation	PIN-P
		Luanta Fir Plantation	LF-P
		Taiwania Plantation	TAI-P
		Japanese Cedar Plantation	JC-P
		Taiwan Incense Cedar Plantation	TIC-P
		Other Conifer Plantation	O-C-P
		Mixed Conifer Plantation	MC-P
		Private Coniferous Plantation	P-C-P
		Newly added Coniferous Plantation	NEW-C-P
Broadleaved	Nature	Natural Broadleaved Forest	B-NF
		Natural Mixed Broadleaved Forest	MB-NF
	Plantation	Acacia Plantation	ACA-P
		Sweet Gum Plantation	SG-P
		Camphor Plantation	CAM-P
		Ash Plantation	ASH-P
		Japanese elm Plantation	JE-P
		Sapphire Dragon Tree Plantation	SDT-P
		Other Broadleaved Plantation	O-B-P
		Mixed Broadleaved Plantation	MB-P
		Private Broadleaved Plantation	P-B-P
		Newly added Broadleaved Plantation	NEW-B-P
Bamboo	Nature	Natural Makino Bamboo Forest	MAK-BAM-NF
		Natural Moso Bamboo Forest	MOS-BAM-NF
		Natural Taiwan Giant Bamboo Forest	TG-BAM-NF
		Natural Thorny Bamboo Forest	THO-BAM-NF
		Natural Bambusa Forest	BAM-BAM-NF
		Natural Fargesia Forest	FAR-BAM-NF
		Other Natural Bamboo Forest	O-BAM-NF
	Plantation	Makino Bamboo Plantation	MAK-BAM-P
		Moso Bamboo Plantation	MOS-BAM-P
		Taiwan Giant Bamboo Plantation	TG-BAM-P
		Thorny Bamboo Plantation	THO-BAM-P
		*Bambusa* Plantation	BAM-BAM-P
		Other Bamboo Plantation	O-BAM-P
		Private Bamboo Plantation	P-BAM-P
Mixed	Nature	Natural Coniferous and Broadleaved Mixed Forest	M-CB-NF
		Natural Bamboo and Coniferous Mixed Forest	M-BAMC-NF
		Natural Bamboo and Broadleaved Mixed Forest	M-BAMB-NF
		Natural Bamboo, Coniferous and Broadleaved Mixed Forest	M-BAMCB-NF
	Plantation	Coniferous and Broadleaved Mixed Plantation	M-CB-P
		Bamboo and Coniferous Mixed Plantation	M-BAMC-P
		Bamboo and Broadleaved Mixed Plantation	M-BAMB-P
		Bamboo, Coniferous and Broadleaved Mixed Plantation	M-BAMCB-P
		Private Coniferous and Broadleaved Mixed Plantation	P-M-CB-P

## Data Availability

Not applicable.

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
