# Peer review of "Modeling of the Spatial Distribution of Forest Carbon Storage in a Tropical/Subtropical Island with Multiple Ecozones"

_plants, 2023, doi:10.3390/plants12152777_

Round 1

Reviewer 1 Report

The research established a model to estimate forest carbon storage and sequestration in the main island of Taiwan considering forest types, ecozones, land use types, stock volume equations, biomass-carbon conversion coefficients, growth and mortality rates.
I found the manuscript well written and easy to read. From my point of view, the methods and the uncertainty analysis are appropriate. I have appreciated, in particular, that the Authors declared all the limitations and uncertainties of data and results. Moreover, the classification of 51 forest types, also taking into account ecozones and land uses demonstrates that the Authors have posed attention on differences among vegetation types as carbon sinks. In my opinion, the research followed a rigorous scientific approach. I also consider the topic relevant in the field.
From my point of view, the manuscript has the standards to be published on Plants - MDPI after minor revision. Here, I highlight my suggestions.

Conclusions section. In my opinion, the Conclusions section is the only part of the text that needs to be improved because it is mainly a summary of the Results and Discussion sections. I think it can be do more incisive by adding information about the usefulness of the model for policymaking, management of natural or semi-natural ecosystems and their services (in particular, helping to make carbon sequestration more efficient in the territory of Taiwan to contribute mitigating global warming), communication of the results to populations and private stakeholders, and other aspects that the Authors can consider relevant. For example, are you planning to promote the use of the model to encourage biodiversity conservation, ecosystem services strengthening and a sustainable use of forests? Please, explain it to give pragmatism to the research.

Other suggestion:

Figure 1. Caption is redundant with the information provided in the legend. I think all the second part of the caption (from “Green area…”) can be removed.

Overall, the manuscript is well written but some sentences need a minor editing.

Examples:

Line 12. Please, change "last to forest" to "last two forest".

Lines 12-13. Maybe the sentence can be reduced to "This study established a model to estimate the distribution of forest carbon storage."

Line 308. Please, change “On” to “on”.

Line 389. Please, change “this” to “This”.

Please, review all the text for minor editing of language.

Reviewer 2 Report

Review of: Modeling of the spatial distribution of forest carbon storage in a tropical/subtropical island with multiple ecozones

Plants

July 2023

Summary:

The authors estimated aboveground biomass in trees on Taiwan to quantify carbon storage in the face of climate change.  They also estimated the increase in carbon capture based on tree growth.  Significant carbon is captured by the trees of Taiwan.

Strengths:

This manuscript is generally well written and the methods are appropriate.  The authors did a great job of dividing the island into ecosystem types and estimating carbon storage both separately for each ecosystem and collectively for Taiwan.

Weaknesses:

There are two large concerns.  First, there was not an attempt to include belowground biomass.  While such measurements are clearly more challenging, there could be considerable carbon stored belowground.  Second, it is not clear if loss of forest land from tree harvest, storms, etc. is included in the overall estimate of carbon capture.  Depending on the frequency and extent of these events, they could play a large role in the total carbon sequestration for Taiwan.

Specific Suggestions:

1.     On line 57, it seems that a year is needed after “in”.

2.     On line 80, we need a source for the fact of how many tree species are on Taiwan.

3.     On line 90, a source is needed for the CSRSR data so that readers can find these data.

4.     On lines 115-116, the authors point out that forest loss from various factors is not included in the estimate.  This seems to be a critical point.  If the authors want to estimate the change in carbon stock, losses must be included in the estimate.

5.     On lines 126-127, the authors point out that belowground carbon was not estimated.  Obviously, estimating belowground carbon is a challenging task.  However, ignoring it completely seems to be a massive oversight.  Are there any allometric equations that would allow for even a rough estimate of belowground biomass (given that there are differences among species, locations, and times of year)?

6.     The assumption on line 174 is a big one.  Can we really assume that there were no significant changes in forest cover if forest loss was not included in the data (as stated on lines 115-116)?

7.     Why are there two y-axes on Figure 4 when there is only one data series?  Probability and frequency are the same pattern multiplied by 10,000.  It seems that one axis will do.

8.     On line 267, replace “in were” with “was”.

9.     On line 268, we need some estimate of how much bigger the carbon pool is when belowground stores are included.  I appreciate the challenge in making this estimate, but ignoring it undermines the purpose of this study.

10.  On page 14, do the increases in carbon storage include forest loss or not?  Lines 115-116 suggest that they do not, which seems to be a critical oversight.

Recommendation:

I recommend that this manuscript be re-reviewed after the authors provide some estimate of belowground carbon storage and clarify if the estimate of annual carbon sequestration does or does not include losses in forest cover along with tree growth. 

Generally well written.  

Round 2

Reviewer 2 Report

The revision is suitable for publication.